# Basal Values of Biochemical and Hematological Parameters in Elite Athletes

**DOI:** 10.3390/ijerph19053059

**Published:** 2022-03-05

**Authors:** Angel Enrique Díaz Martínez, María José Alcaide Martín, Marcela González-Gross

**Affiliations:** 1ImFINE Research Group, Department of Health and Human Performance, Facultad de Ciencias de la Actividad Física y del Deporte-INEF, Universidad Politécnica de Madrid, 28040 Madrid, Spain; marcela.gonzalez.gross@upm.es; 2Clinical Laboratory Unit, Department of Sport and Health, Spanish Agency for Health Protection in Sport (Agencia Española de Protección de la Salud en el Deporte—AEPSAD), 28040 Madrid, Spain; 3Department of Laboratory Emergency, La Paz University Hospital, 28046 Madrid, Spain; mjose.alcaide@salud.madrid.org; 4Centro de Investigación Biomédica en Red de Fisiopatología de la Obesidad y Nutrición (CIBEROBN), Instituto de Salud Carlos III, 28040 Madrid, Spain

**Keywords:** biochemistry, hematology, clinical laboratory, exercise, athletes

## Abstract

The purpose of this study was to show how continuous exercise affects the basal values of biochemical and hematological parameters in elite athletes. A total of 14,010 samples (male = 8452 and female = 5558 (March 2011–March 2020)) from 3588 elite athletes (male = 2258 and female = 1330, mean age 24.9 ± 6.9 vs. 24.1 ± 5.5 years, respectively) from 32 sport modalities, were studied over 9 years to check the variation of basal biochemical and hematological parameter values. There were differences seen in the basal values of creatine kinase (CK), urea, creatinine, aspartate transaminase (AST), alanine aminotransferase (ALT), lactate dehydrogenase (LDH), potassium, total bilirubin, and eosinophil percentage compared to reference population data. However, other analytes showed narrow ranges of variation like glucose, total protein, albumin, sodium, hemoglobin, mean cell volume (MCV), and platelet count. Exercise produces changes in biochemical and hematological basal values of athletes compared to the general population, with the greatest variation in CK, but AST, ALT, LDH, potassium, and total bilirubin (TBil) show high values in serum, only with a wider distribution of values. The data here reflects the effect of exercise on biochemical and hematological parameter baseline ranges in elite athletes. As clinical laboratories use reference intervals to validate clinical reports, these “pseudo” reference intervals should be used when validating laboratory reports.

## 1. Introduction

Regular physical exercise and training induces physiological and metabolic adaptations [1,2,3,4]; all organs and systems are affected and sports performance is linked to such adaptations. For trainers, sports physicians, and athletes it is important to monitor these adaptations. Blood biomarkers have been proposed as adequate markers for measuring the training effect in the short and long term, but also for maintaining health, identifying chronic stress, inflammation, fatigue, or as prevention of injuries [4,5,6].

As health check control, athletes regularly undergo analytical controls that include biochemical and hematological parameters to ensure a correct state of health and the absence of any abnormality that may diminish exercise performance. These parameters include glucose, urea, creatinine, transaminases, creatine kinase, ions, iron, etc., as well as erythrocyte count and associated parameters, total leukocytes and leukocyte subpopulations, platelets, and sometimes reticulocytes. However, continuous and intense exercise, training, and competitions can produce changes in many blood parameters, which should not be considered as pathological but as induced by exercise [7,8]. Fallon [9] performed a screening of biochemical parameters in 100 elite athletes from 11 sports. The athletes were involved in training once or twice a day for at least 6 days per week. A total of 194 abnormal results were found in 82 of the athletes. The most common abnormalities were increases in aspartate transaminase (AST) (27%), phosphate (13%), creatine kinase (CK) (13%), urea (12%), and bilirubin (12%).

Lippi [10] compared several blood biomarkers among 80 professional cyclists, 37 cross-country skiers, and 60 male healthy sedentary controls. Results showed that values of laboratory testing lay outside conventional reference limits and might express physiological adaptations to regular physical aerobic exercise, emphasizing the need for the estimation of reliable reference limits in elite and professional athletes.

Kratz [11] stated in 2002 that using reference intervals (RI) for the general population might not be appropriate for marathon runners, and that it would be desirable to establish specific RIs for these athletes. In their study, they proposed RIs calculated with 37 marathon runners before running a marathon, immediately after the marathon, and 24 h after the marathon, recommending the non-parametric method to calculate RIs.

Several studies have aimed to establish RIs for single parameter in elite athletes, i.e., reticulocyte count and associated parameters [12] in football, rugby players, and skiers; CK [13] in football and swimming; CK [14] in football; LDH [15] in cycling, long and middle distance running, rowing, sprint, swimming, water polo, hockey, and wushu; and Creatinine [16] in rugby players, soccer players, skiers, sailors, and cyclists; however large-scale studies of RIs on elite athletes are missing [5].

Analysis of biochemical and hematological data in elite athletes requires caution, because of an intriguing issue in sports medicine: the identification and appropriate implementation of references ranges in laboratory tests for athletes [16]. It is difficult to establish true resting values in elite athletes who train for several hours every day [17].

Values from laboratory tests that lie outside conventional reference limits—calculated using values from sedentary populations—may reflect physiological adaptations to regular and demanding physical aerobic activity than resulting from pathologies [16]. Thus, the RIs for the general population are not be valid for elite athletes [17].

The newly revised document, *Defining, Establishing, and Verifying Reference Intervals in the Clinical Laboratory: Approved Guideline-Third Edition (C28-A3)*, published in November 2008, helps define the criteria for the laboratories for selecting a healthy reference population, determine how many subjects are needed, identify outliers, and perform the calculation to generate a valid RI [18].

An RI is defined as the range of values for a clinical test, where 95% of the healthy population is included [19,20], centered on the median, and describes a specific population [21]. It is an arbitrary but common decision to define the RI as the central interval that comprises 95% of the data, surrounded by the 2.5 and 97.5 percentiles [15,22,23,24,25].

RI should be conducted with data from healthy subjects (noting that health is a relative condition lacking a universal definition [20]) in standardized and controlled situations (general population). Notably, an elite athlete (a subgroup of the general population that includes healthy persons that do a high intensity, amount (1–2 times per day), and quality exercise) maintains their training regime and competitions whether they have planned a blood draw for analytical control [9].

Some researchers favor the indirect method, also known as data mining, based on previous laboratory results to obtain RIs [20]. The IFCC Committee on Reference Intervals and Decision Limits (IFCC C-RIDL) aims to encourage using indirect methods to establish and verify Ris and to promote the publication of such intervals with clear explanations of the process used [26].

Clinicians compare the values reported by laboratory professionals with the RIs. Whether a test is normal or abnormal is perhaps the most important decision related to a laboratory test [18].

Good laboratory practice states that each laboratory should perform its own RI values for their methods and their population of subjects [19]. According to the ISO 15189 standard, for clinical laboratory accreditation, it states that each laboratory should periodically re-evaluate its RIs [27]. Usually, RIs for women, men, and children may be used, but it is impossible to use specific RIs for sports with different types of exercise when validating clinical laboratory reports. Therefore, the aim of this study was to evaluate how continuous exercise (from different sports federations and stages of exercise), affects basal values of biochemical and hematological parameters in elite athletes, to compare the results obtained with those established for the general population and to propose a basal values range (Pseudo RIs), to use during clinical report validation.

The clinical laboratory of Agencia Española de Protección de la Salud en el Deporte (AEPSAD) used unique basal values range (Pseudo RIs) for all the sports federations (obtained in 2005–2008 with 7044 samples) at the time of clinical validation of laboratory reports, but data presented in this study (2011–2020 cohort) can provide an update.

## 2. Materials and Methods

### 2.1. Subjects

This study was performed at the clinical laboratory of the Sports Medicine Center of the Spanish Agency for Health Protection in Sport (AEPSAD) in Madrid. This center is the only one with a state character in Spain and carries out sports medical examinations of high-level Spanish athletes. The clinical laboratory performs biochemical, hematological, and hormonal analysis on blood and urinary samples obtained from elite athletes.

Data were obtained between March 2011 and March 2020, from the clinical laboratory database, Modulab gold version 2.0 (Werfen Spain, L’Hospitalet de Llobregat, Barcelona. Spain). A total of 14,010 samples (male = 8452 and female = 5558; mean age 24.9 ± 6.9) from 3588 elite athletes (male = 2258 and female = 1330; mean age 24.1 ± 5.5), were studied to evaluate the intervals of variation of baseline values for biochemical and hematological parameters.

All the athletes in this study went to the clinical laboratory of the Sports Medicine Center, either for medical examination or periodic control of blood tests. Before medical examination and blood extraction, all athletes had given their informed consent to carry out the clinical tests, and to further use the data anonymously for scientific research. Sex, age, race, sport federation, biochemical, and hematological data were only collected to obtain the basal values to protect anonymity.

All subjects were elite athletes from different sports (Appendix A). No samples analyzed in this study were post-exercise samples. All blood samples were obtained after a night’s rest and in a fasting state.

All the samples were analyzed using the same analytical equipment and reagents, independently of the sport federation and the training or competition period. Appointments for the clinical laboratory were made by the doctors of their respective sports federations. Consider this, we obtained the data limits of baseline variation for 22 biochemical parameters and 32 hematological parameters, in elite athletes during 2011–2020.

Athletes over 18 years were selected for the study. The largest group of samples (97.35%) were from athletes aged 18 to 40. Blood samples from athletes over 40 years comprised 2.65% and athletes over 50 years comprised 0.72%.

The racial distribution of blood samples was: Caucasian (95.04%), African American (2.72%), Arab (1.14%), Asian (0.68%), and South American (0.42%).

None of the female athletes who participated in this study were pregnant at the time of blood collection.

Reference intervals for the general population have been obtained from the manufacturers of the analyzers, reagents, calibrators, and controls used: Beckman Coulter SLU (biochemical parameters) and Siemens Healthineers (hematological parameters).

### 2.2. Methods

The blood sampling procedure was maintained throughout the cohort period to minimize pre-analytical variation. Fasting blood samples were obtained by puncture using the aseptic technique of a vein in the cubital fossa between 9:00 a.m. and 10:00 a.m.

Vacutainer tubes (3 mL), with EDTA-K3 as anticoagulant, were used for the hematological study, ref. 388630, (Beckton, Dickinson and Company, Franklin Lakes, NY, USA). Whereas 8.5- or 5-mL vacutainer tubes with clot activator and separator gel were used for the biochemical study, ref. 367953 and ref. 367955, respectively (Beckton, Dickinson and Company).

Tubes for biochemical studies were kept for at least 20 min, to allow clot formation. Afterwards, tubes were centrifuged at 4000 rpm for 10 min, at 20 °C. Biochemical parameters were analyzed on a Beckman Coulter AU400 analyzer (Beckman Coulter Inc., Brea, CA, USA), and hematological parameters were analyzed on an Advia 120 analyzer (Siemens Healthneers, Erlangen, Germany). For each instrument, reagents, calibrators, and controls were from the same manufacturer as the respective analyzer. Instrument operation and calibration followed manufacturer’s instructions. All analyses were performed on the day of blood extraction. Biochemical parameters are shown in Appendix A and hematological parameters are shown in Appendix A.

The Estimated Glomerular Filtration Rate (eGFR) was calculated using the Modification of Diet in Renal Disease (MDRD) and Chronic Kidney Disease Epidemiology Collaboration (CKD-EPI) formula.

The methods of analysis have not changed over the time of study. Only change of lots of the different reagents, calibrators, and quality controls have occurred.

### 2.3. Statistical Analysis

Statistical analysis was performed with Reference Value Advisor v2.1 [28] on Excel 2013 (Microsoft, Redmond, Washington, DC, USA). The software identifies outlier data and shows the parametric untransformed data, Box-Cox transformed data, non-parametric RI (percentiles 2.5–97.5), and 90% CI for the lower and upper limits.

T-test were used to compare the means of the 2011–2020 group with data of the general population, and to compare women and men means of the 2011–2020 group of biochemical and hematological parameters.

## 3. Results

The biochemical parameters urea, creatinine, CK, AST, ALT, LDH, potassium, and total bilirubin (TBil) showed greater variation of basal values compared to those for general population, whereas other analytes—glucose, total protein, albumin, and sodium—showed narrower intervals of variation (Table 1). eGFR calculated using MDRD 2009 and CKD-EPI formula are shown in Table 2.

Results for urea showed a higher upper limit compared to the RI for the general population (Table 1).

Calcium and magnesium showed similar RIs when compared to the general population; however, a slight increase in the lower limit for calcium and a slight decrease in the upper limit for magnesium were found in elite athletes. The lower and upper limits of phosphate basal values are higher than those observed in the general population. The data found for serum iron RI in elite athletes showed a marked decrease in the lower limit in both women and men and a slight increase in the upper reference limit. (Table 1).

Regarding hematological parameters, the upper limit for hemoglobin and mean cell hemoglobin (MCH) are slightly lower and markedly lower in mean cell volume (MCV) than those found for the general population, whereas the interval of basal values for erythrocytes, hematocrit, and mean cell hemoglobin concentration (MCHC) are like those described for the general population. Red cell distribution width (RDW) showed a narrow RI to those described for the general population (Table 3).

The basal values found for reticulocyte count, reticulocyte percentage and HCr were like those found as RIs for the general population, with MVCr and fractions of reticulocyte populations (RETH, RETM, RETL) being different from those found for the general population. (Table 4).

Baseline values for total leukocyte count was lower than the general population. Furthermore, lymphocyte, neutrophils, monocytes, eosinophils, basophils, and large unstained cells (LUC) upper limit counts were lower than the general population. According to the percentage count, lymphocytes and eosinophils showed upper limits of the baseline values, higher than those for the general population, with neutrophils showing a marked decrease at both baseline values limits, and monocytes showing a marked increase at the lower limit but a decrease at the upper limit of baseline values. Basophils and LUC showed lower data than those described for the general population (Table 5).

The platelet count data found showed a marked decrease in the upper limit of baseline values compared to the RI of the general population, but an increase in the mean platelet volume (MPV) was found. The baseline values for plaquetocrit (Table 6) showed a marked decrease in the upper limit of baseline values, compared to general population RIs.

## 4. Discussion

### 4.1. Biochemical Parameters

#### 4.1.1. Markers of Muscle Injury

The greatest variation in basal values regarding the reference range for the general population was found in CK activity. Physical exercise causes rhabdomyolysis with the liberation of the content of the muscular cell, and an increase in serum levels of skeletal muscle enzymes (CK, AST, ALT, LDH), which are an index of tissue damage following acute or chronic injuries [14,29,30,31]. Total CK levels depend on age, gender, race, muscle mass, physical activity, and climatic condition [14,29,30]. Its clearance from plasma depends on the training level, type, and intensity as well as the duration of exercise [14].

CK activity shows a great variation, including individual variation in CK response to damaging exercise [32]. Moreover, serum CK activity measured in individuals exercising to a similar degree showed high variability [14]. In our study, CK showed much higher baseline values than the RIs for the general population, consistent with previously published literature in male football (soccer) players and swimmers [13]. Nunes et al. obtained an upper limit for the 97.5% percentile for CK activity of 1338 (UI/L) (CI: 1191–1639 UI/L) with 128 male professional soccer players, similar to results in this study. Nunes et al., suggests a 90% percentile (975 UI/L) upper plasma CK limit for the early detection of muscle overload for competing soccer players [14].

Even if it is typical for athletes to have elevated CK during training, suggested reference ranges of 82–1083 UI/L in male athletes and 47–513 UI/L in female athletes [32] are lower than values of our study.

In our study, AST and ALT showed a higher upper limit compared to the RIs in the general population. The elevated AST and ALT activity in athletes, excluding other pathologies, should be considered of muscular and not hepatic origin [7,8,33]. The baseline values for LDH in this study showed an upper limit higher than the RIs described for the general population. Exercise triggers transient elevations of muscular and cardiac biomarkers, including CK; AST; ALT; and LDH, among others [7,8,11,29,30,31,33].

LDH enzyme activities are found in every tissue, with its highest activity in the skeletal muscle, liver, heart, kidney, brain, lungs, and erythrocytes. LDH enzyme activity in serum is a biochemical marker for muscular damage [15]. LDH RIs were obtained in 320 male and 252 female athletes from 10 sports, including cycling, long distance running, rowing, sprint, swimming, hockey, and wushu. Furthermore, the authors proposed a general RI for LDH for a total of athletes [15], and showed that the mean LDH activity value is higher in male athletes compared to female athletes for all the sports [15]. In this study, a higher upper limit of LDH, in both females and males, was found when compared to the general population.

#### 4.1.2. Substrates

Data found for serum glucose basal values showed a slight decrease in both limits. In the same way, Lippi et al. showed a decrease in the mean glucose concentration in professional skiers and cyclists compared to male sedentary healthy controls. The lower serum glucose concentration in athletes could be explained by better metabolic control, especially in terms of improved glucose tolerance and insulin sensitivity, common features of regular aerobic physical activity [10].

The basal values for serum creatinine showed a wider range of data, with higher values at the upper limit, compared to those found for the general population. However, the reference values commonly used for athletes are those defined for normal, sedentary people [16].

Although personal uncontrolled use of creatine by single athletes cannot be excluded, the large number of subjects recruited from all the sports should minimize the possible influence of creatine use. Furthermore, creatine supplementation has a minimal effect on creatinine concentrations and renal function in young healthy adults, as reported in a review of literature from 1966 to 2004 [16].

Creatinine is therefore a fairly stable variable in athletes, but its concentration may differ from those of sedentary people and sometimes among sports. The serum creatinine concentrations found in professional athletes can vary according to their modality, the training load, aerobic/anaerobic metabolism, the lengths of their competitions, and at different stages of a competitive season in the same athletes [14,16]. Differences between sports disciplines can be linked to BMI. Banfi found a correlation between creatinine and BMI in the overall group of athletes studied [16]. Furthermore, Lippi found lower mean serum creatinine values in professional cross-country skiers and cyclists compared to male sedentary healthy controls [10].

Results obtained in the AEPSAD clinical laboratory have shown a lower serum creatinine concentration in child and disabled athletes (e.g., wheelchair and limb amputees), sometimes below the lower reference limit, which could be due to decreased muscle mass (unpublished data).

The concentration of creatinine in serum has long been widely used and commonly accepted measure of renal function in clinical medicine [14,16]. The MDRD study equation was developed in people with CKD, therefore, its major limitations are imprecision and systematic underestimation of measured glomerular filtration rate (GFR) (because of bias) at higher levels [34].

The MDRD has been used since its development, but more recent formula—CDK-EPI—is being recommended because it is more accurate across a wide variety of populations and clinical conditions [34]. Therefore, the National Kidney Foundation has recommended that clinical laboratories should use the CKD-EPI equation to report estimated GFR. Data obtained in the eGFR 2011–2020 group, obtained with the CKD-EPI formula, showed higher upper limits than the eGFR obtained with the MDRD formula (Table 2). Serum creatinine RI in athletes are higher than that described for the general population, so an underestimate of eGFR in athletes can be obtained. Using the 2012 CKD-EPI cystatin C equation is as accurate as the 2009 CKD-EPI creatinine equation at estimating eGFR, and does not require specification of race; further, it is more accurate in subjects with high creatinine levels [35], such as athletes.

Biomarkers such as serum creatinine and urea, are also used to monitor the effects of training [14]. Urea is the end product of the degradation of nitrogenous compounds from proteins, and it is synthesized in the liver and excreted by the kidneys. The serum concentrations of urea have been used as a marker of protein catabolism and stimulate gluconeogenesis during higher training loads. Serum urea nitrogen can indicate overall protein synthesis vs. breakdown [32].

Urea levels found in 2011–2020 group, in both females and males, showed an increase in both the lower and upper limits of the basal values in elite athletes compared to the RIs for the general population. This indicates the need for appropriated reference ranges for laboratory tests in athletes, as indicated for creatinine [16].

Higher urea nitrogen may be due to exhaustive exercise training, catabolism, and high dietary protein intake [32]. Some doctors and trainers use serum levels of urea and CK activity to check the volume (duration) and intensity of the exercise. It was proposed that monitoring the serum urea concentrations and the CK activity may indicate an acute impairment in exercise tolerance [14].

Because dehydration has a negative impact on kidney function, the ratio of urea to creatinine has been a strong indicator of hydration state with a suggested threshold of 20 for dehydration [32]. Both blood osmolality and sodium levels have been used for hydration assessment because both values increase linearly with the levels of dehydration. Most studies suggest that the threshold of dehydration for blood osmolality is 295 mmol·kg^−1^ of plasma water [32]. Even a little dehydration (e.g., −1% of body weight) can notably increase plasma osmolality [32]. Serum osmolality can be estimated by the measurement of glucose, urea, and sodium in serum.

In this study, total protein and albumin showed a narrower range compared to RIs of the general population. Furthermore, a small decrease in the mean value of serum albumin were also found in professional skiers and cyclists compared to male sedentary healthy controls [10].

An imbalance between dietary protein intake and dietary protein needs may cause net protein loss in athletes. A combination of biomarkers including total protein, albumin, and urea may help athletes to gauge their protein status and make dietary changes to improve training outcomes [32]. Albumin has been associated with human growth hormone concentrations in the blood, although the mechanism by which these two markers are related is unknown [32]. In the absence of disease, low blood protein, low albumin, and elevated urea may indicate insufficient protein intake in athletes [32].

Uric acid is the final metabolite of purine nucleotides (ATP and GTP), with ATP being used for muscular contraction during exercise. This study showed an increase in both limits of basal values.

Athletes and physical active subjects displayed an enhanced antioxidant capacity with increased serum concentrations of uric acid. Nunes et al. found a slightly higher serum uric acid RI in athletes (0.24–0.49 mmol/L) than in sedentary subjects (0.23–0.47 mmol/L), and this difference can be explained by the intensity of the subject’s training [14]. However, a decrease in the mean serum uric acid value were found in professional skiers and cyclists compared to male sedentary healthy controls [10].

Physical exercise causes both hemolysis and rhabdomyolysis [36,37,38,39], which can increase serum LDH activity, potassium, and TBil concentration.

Data obtained for TBil showed an upper limit, higher than that described for the general population. In athletes, the main cause of increased serum bilirubin is hemolysis and subsequent catabolism of hemoglobin. There is a wide body of literature reporting red cell hemolysis occurring after various forms of exercise. Telford et al. indicated that, whereas general circulatory trauma to the red blood cells associated with 1 h of exercise at 75% maximal oxygen uptake may cause some exercise-induced hemolysis, trauma associated with foot strike is the major contributor to hemolysis during running [8,38]. In addition, we found elevations in serum TBil concentration in combat sports (boxing, karate, and judo) and in team sports, such as handball and basketball, due to bruising that may be caused during combat or defense systems (unpublished data).

#### 4.1.3. Ions

Data obtained in our study for serum sodium showed an upper limit lower than that described for the general population, while data obtained for potassium showed an upper limit higher than that described for the general population. Basal values for chloride showed a decrease in both limits compared to the RIs described for the general population.

Calcium and magnesium showed similar basal values when compared to the general population; although, a slight increase in the lower limit for calcium and a slight decrease in the upper limit for magnesium were found in elite athletes. Magnesium is a ubiquitous element that plays a fundamental role in many cellular reactions. Over 300 metabolic reactions require magnesium as a cofactor. Sports related causes of hypermagnesemia include rhabdomyolysis and the ingestion of magnesium containing substances such as vitamins, anti-acids, or cathartics. Hypermagnesemia is rare compared to hypomagnesemia. In athletes, hypomagnesemia is possibly caused by excessive sweating while training. Magnesium levels return to normal values within 24 h after exercise [17,36].

The lower and upper limits of phosphate basal values found in our study were higher than those observed in the general population.

Phosphate plays an important role in different biochemical reactions, mostly related to energy production. Trauma, rhabdomyolysis, and use of nutritional supplements are sports-related causes of hyperphosphatemia [17]. Malliaropoulos et al. found that 47% of the athletes in their study presented a mild increase in serum phosphate, with most phosphate and magnesium serum alterations close to the upper limits of a normal non -athletic population, indicating that the RIs for the general population may not be valid for elite athletes [17].

A significant decrease in the lower limit of the basal serum iron value has been obtained. Although iron deficiency is most common in female athletes (15–35% athlete cohorts deficient), approximately 5–11% of male athlete cohorts also present with this issue [40].

Iron concentration reflects total iron content. Between and within-day variation of iron concentration is high (10–26%), therefore, iron concentration must be interpreted cautiously and cannot be rendered a useful measure of iron status alone [32]. Iron homeostasis is regulated by hepcidin, an antimicrobial peptide synthesized in the liver. Hepcidin binds, internalizes, and degrades ferroportin—the cellular iron exporter—blocking the absorption of dietary iron into the circulation, inhibiting iron recycling macrophages and the release of body iron stores. Regular exercise, especially endurance training, has been associated with iron deficiency and iron deficiency anemia. Exercise is also known to increase inflammation-responsive cytokine levels, especially that of interleukin-6 (IL-6), a major regulator of hepcidin expression [41].

### 4.2. Hematological Parameters

In this study, upper limits for hemoglobin, MCH, and MCV are slightly lower, whereas the rest of red blood cells (RBC) associated parameters were like those of the general population. Exercise causes an increase in RBC count, hemoglobin concentration, and hematocrit due to hemoconcentration [11,42]. However, although exercise causes an increase in RBC count, it is common to find hemodilution in samples taken 24 h post-exercise [11,36], which does not return to baseline values until about 72 h after exercise [36]. Exercise modalities were found to have important effects on hematologic parameters.

Basal values found for reticulocyte count were like the RIs found for the general population. Banfi et al. studied the reticulocyte count and associated parameters, using a Coulter LH700 instrument, in 106 elite athletes, making a proposal of reference values [12]. The RIs found in elite athletes—football and rugby players, and skiers—were compared with data found in a control group of 73 men (general population), showing results without statistically significant differences with the control group. Reticulocyte distribution in athletes was like that found in the general population (RI of reticulocytes was 0.30–1.54%, and the mean reticulocyte volume was 93.1–114.8 fL). There are differences between automated systems, particularly when different dyes and/or marking methods are used to identify reticulocytes [12], which may explain the difference in the data found in our study, showing a higher RI for % reticulocytes, at both limits, but a similar mean reticulocyte cell volume (MCVr). Furthermore, authors showed that there were no observed changes in the reticulocyte manual count in 20 trained athletes before, during, and after completing a 42.2-km race; reticulocytes remained inside the normal range [43].

In the 2011–2020 group, a decrease was found at both basal values limits for the total leukocyte count, and a decrease in lymphocyte, neutrophil, monocyte, eosinophil, basophil, and LUC counts, mainly at the upper limit of the RI, compared to those described for the general population. According to the percentage count, the upper limit of eosinophils is striking, showing a value much higher than that of the RI established for the general population. This could be due to the augmented amount of air inspired, and the exposure to air antigens. Short and intense exercise leukocytosis is mostly due to an increase of lymphocyte and neutrophils, because of epinephrine levels. However, long distance exercise (marathon) leukocytosis is mostly due to the striking increase in neutrophils count, with lymphocytes and eosinophils counts showing a marked decrease mainly because of cortisol [33,36].

Exercise can induce an elevation of leukocyte count, mainly because of demargination of WBCs secondary to increased blood flow and exercise-induced increases in epinephrine and cortisol levels [33]. However, under baseline conditions it is common to find low leukocyte counts, sometimes close to leukopenia [7,44,45].

Although the WBCs assay cannot be used alone to assess an athlete’s level of inflammation, it provides valuable information about shifts in immune cell populations that may occur during muscle damage-induced inflammation. Another benefit of assessing WBCs profiles in athletes is that they can diagnose potential inflammation-causing infections or diseases and increases in biomarkers that are common with muscle damage-induced inflammation [32].

The platelet count found in both groups showed a marked decrease in the upper limit of RI, compared to the RI of the general population. The authors have shown that exercise can induce an elevation in the platelets count of about 30% [7,36], and platelet aggregation is enhanced during moderate exercise [33], thus this decrease in the upper limit in basal state may represent an adaptation to prevent thrombotic situations.

Clinical laboratories use RIs to validate clinical reports. Usually, a RI for women, men, and children may be used, but it is impossible to use specific RIs for different sports with different types of exercise, at the time of laboratory reports validation. Moreover, it is difficult to establish true resting values in elite athletes who train for several hours every day. New reference limits should be established for biochemical variables in elite athletes, whose hydration, nutritional, and fasting status are difficult to ascertain, and probably at variance from the normal non-athletic population. The RIs for the general population are not valid for elite athletes. Exercise is a well-known source of pre-analytical variation [33], but as the patients in the clinical laboratory of the Centre for Sports Medicine are elite athletes, and elite athletes do not stop training or competing for blood sampling, therefore, the values found may represent their actual ongoing situation and not an artificial or altered situation [9].

The aims of this study were to evaluate how continuous exercise (from several years, from different sports federations), affects basal values of biochemical and hematological parameters in elite athletes, and to propose a basal values range to be used at the validation time of clinical reports at the clinical laboratory.

With these data (2011–2020) and other data obtained previously (2005–2008), the coefficients of intraindividual and interindividual variation (CVW and CVB) have been calculated, which together with the coefficient of analytical variation (CVA), allow us to calculate the Reference Change Value (RCV) that allows us to compare the last result with the previous one, and to see if the variation found is due to analytical and biological factors or if other factors have intervened. Values outside the proposed limits should be checked and, if confirmed, physicians should be advised to follow up the athlete.

## 5. Conclusions

The RIs for the general population are not valid for elite athletes.

New reference limits should be established for biochemical variables in elite athletes, whose hydration, nutritional, and fasting status are difficult to ascertain, and most likely at variance from the normal non-athletic population.

With this study, we propose a basal values range to be used at the validation time of clinical reports at the clinical laboratory.

Using these data of basal values variation and data of Reference Change Value (RCV) we can check the last result with the previous results and see if the variation found is due to analytical and biological factors or if other factors have been implicated. Values outside of the limits proposed should be checked and, if confirmed, physicians should be advised to follow up the athlete.

### Limitations

There are no established protocols for the optimal timing of blood sampling in elite athletes, and the timing may be different in different sports, due to the training and competition schedule between sports. The clinical laboratory only extracted blood samples and analyzed them. Schedules of blood control were made by the doctors of their respective sport federation; thus, all samples were analyzed independently of the time of the year (training period or competition) and of the sport federation.

There are limitations regarding physical fitness indices, anthropometric measurements, or dietary intake, and no data are included in this article.

Authors recognize that differences may occur among different types of exercise (strength or endurance), but this approach shows how continuous exercise affects basal values of biochemical and hematological parameters, thus these data could be used as “Pseudo RIs” at the time of clinical validation of laboratory reports.

## Figures and Tables

**Table 1 ijerph-19-03059-t001:** Reference intervals (RIs) and basal values for biochemical parameters.

Parameter	Units	Number	Female	Female 2011–2020 90% CI		Female General Population RI	Number	Male	Male 2011–2020 90% CI		Male General Population RI
			2011–2020	Lower Limit	Upper Limit			2011–2020	Lower Limit	Upper Limit	
Gluc	mmol/L	13,929	3.98–5.72 **	3.94–4.0	5.72–5.77	4.11–5.89	13,929	3.98–5.72 **	3.94–4.0	5.72–5.77	4.11–5.89
Urea	mmol/L	5550	3.56–9.16 **^&&^	3.50–3.61	9.07–9.26	2.83–7.16	8396	4.21–10.11 **^&&^	4.15–4.28	9.97–10.21	3.20–8.20
Crea	μmol/L	5550	67.19–107.85 **^&&^	66.30–67.19	106.97–108.73	58.35–96.36	8396	76.03–125.53 **^&&^	75.14–76.03	124.65–126.41	74.26–110.50
UA	mmol/L	5550	0.167–0.381 *^&&^	0.167–0.173	0.375–0.387	0.155–0.357	8393	0.220–0.470 **^&&^	0.214–0.220	0.470–0.476	0.208–0.428
Chol	mmol/L	13,943	3.16–6.44	3.13–3.18	6.39–6.52	<5.17	13,943	3.16–6.44	3.13–3.18	6.39–6.52	<5.17
TG	mmol/L	13,943	0.37–1.67	0.37–0.38	1.63–1.71	<1.69	13,943	0.37–1.67	0.37–0.38	1.63–1.71	<1.69
TP	g/L	13,942	65–80 **	65–65	80–80	66–83	13,942	65–80 **	65–65	80–80	66–83
Alb	g/L	13,942	40–50 **	39–40	50–50	35–52	13,942	40–50 **	39–40	50–50	35–52
CK	U/L	5562	59.0–785.9 **^&&^	58.0–61.1	732.3–826.9	35–210	8423	87.0–1476.4 **^&&^	85.00–89.00	1394.0–1566.4	50–400
AST	U/L	5550	16.0.–54.0 **^&&^	16.0–16.0	52.0–56.0	<35	8397	18.0–69.0 **^&&^	18.00–18.00	68.00–72.00	<50
ALT	U/L	5550	10.0–43.0 **^&&^	10.0–10.0	41.0–44.0	<35	8397	12.0–55.0 **^&&^	12.00–12.00	53.00–57.00	<35
GGT	U/L	13,492	9.0–42.0 **	9.0–9.0	40.7–43.0	<50	13,492	9.0–42.0 **	9.0–9.0	40.7–43.0	<50
LDH	U/L	13,947	136.0–284.0 **	135.0–136.0	282.0–287.0	<248	13,947	136.0–284.0 **	135.0–136.0	282.0–287.0	<248
Na	mmol/L	13,922	136.0–142.0 **	136.0–136.0	142.0–142.0	136–146	13,922	136.0–142.0 **	136.0–136.0	142.0–142.0	136–146
K	mmol/L	13,918	3.9–5.4 **	3.9–3.9	5.4–5.4	3.5–5.1	13,918	3.9–5.4 **	3.9–3.9	5.4–5.4	3.5–5.1
Cl	mmol/L	13,926	99.4–107.1 **	99.3–99.5	107.0–107.2	101–109	13,926	99.4–107.1 **	99.3–99.5	107.0–107.2	101–109
Mg	mmol/L	12,806	0.74–0.99 **	0.74–0.74	0.95–0.99	0.74–1.03	12,806	0.74–0.99 **	0.74–0.74	0.95–0.99	0.74–1.03
Ca	mmol/L	13,938	2.27–2.64 ^NS^	2.27–2.27	2.62–2.64	2.20–2.64	13,938	2.27–2.64 ^NS^	2.27–2.27	2.62–2.64	2.20–2.64
Phos	mmol/L	13,942	0.94–1.58 **	0.94–0.94	1.58–1.58	0.81–1.45	13,942	0.94–1.58 **	0.94–0.94	1.58–1.58	0.81–1.45
Fe	μmol/L	13,965	5.53–34.17 **^&&^	5.30–5.82	33.68–35.12	10.74–32.23	13,965	7.70–34.79 **^&&^	7.54–7.86	34.15–35.42	12.53–32.23
TBil	μmol/L	13,912	5.13–25.66 **	5.13–5.13	25.66–25.66	3.42–22.24	13,912	5.13–25.66 **	5.13–5.13	25.66–25.66	3.42–22.24

See ***Abbreviations and symbols*** section for the meaning of the abbreviations. Differences of means between the 2011–2020 group and general population RI (* = *p* < 0.05, ** = *p* < 0.01). Differences of means between women and men in the 2011–2020 group (^&&^ = *p* < 0.01). ^NS^: No statistically significant difference.

**Table 2 ijerph-19-03059-t002:** eGFR estimation in females and males using MDRD (2009) and CKD-EPI formulas.

	N	MDRD (2009)			CKD-EPI		
		Mean (SD)	LL RI	UL RI	Mean (SD)	LL RI	UL RI
Female	5558	72.0 (10.8)	54.1	96.2	75.3 (14.8)	51.9	108.5
Male	8452	82.0 (13.0)	60.4	110.1	87.2 (19.1)	56.3	127.7

LL RI: Lower limit of Reference Interval. UL RI: Upper limit of Reference Interval.

**Table 3 ijerph-19-03059-t003:** Reference intervals (RIs) and basal values for red blood cells and associated parameters.

Parameter	Units	Number	Female 2011–2020	Female 2011–2020 90% CI		Female General Population RIs	Number	Male 2011–2020	Male 2011–2020 90% CI		Male General Population RIs
				Lower Limit	UPPER LIMIT				Lower Limit	Upper Limit	
RBC	10^12^/L	4763	3.99–5.18 ^NS&&^	3.96–4.00	5.16–5.21	3.9–5.20	7225	4.52–5.84 **^&&^	4.50–4.53	5.82–5.86	4.3–5.75
Hb	g/L	4763	119–151 **^&&^	118.0–119.1	151.0–151.0	120.0–156.0	7225	135.0–170.0 ^NS&&^	135.0–136.0	170.0–171.0	135.0–172.0
MCV	fL	11,988	80.9–95.0 **	80.7–81.1	94.8–95.1	80–99	11,988	80.9–95.0 **	80.7–81.1	94.8–95.1	80–99
Hct	L/L	4763	0.356–0.452 ^NS&&^	0.354–0.358	0.450–0.454	0.355–0.450	7255	0.400–0.504 ^NS&&^	0.399–0.402	0.503–0.505	0.395–0.505
MCH	pg	11,988	26.6–32.3 **	26.4–26.7	32.2–32.3	27–33.5	11,988	26.6–32.3 **	26.4–26.7	32.2–32.3	27.0–33.5
MCHC	g/L	11,975	315–361 ^NS^	314–316	361–362	315–360	11,975	315–361 ^NS^	314–316	361–362	315–360
RDW	%	11,988	12.0–14.2 ^NS^	12.0–12.0	14.2–14.3	11.5–14.7	11,988	12.0–14.2 ^NS^	12.0–12.0	14.2–14.3	11.5–14.7

See ***Abbreviations and symbols*** section for the meaning of the abbreviations. Differences of means between the 2011–2020 group and general population RI (** = *p* < 0.01). Differences of means between women and men in the 2011–2020 group (^&&^ = *p* < 0.01). ^NS^: No statistically significant difference.

**Table 4 ijerph-19-03059-t004:** Reference intervals (Ris) and basal values for reticulocyte count and associated parameters (females and males).

Parameter	Units	Number	2011–2020	2011–2020 90% CI		General Population Ris
				Lower Limit	Upper Limit	
Rtc	10^9^/L	5594	30.30–98.01 ^NS^	29.80–30.99	96.40–99.20	25–100
Rtc%	%	5594	0.60–2.01 ^NS^	0.59–0.62	2.00–2.05	0.5–2
HCr	pg	5594	28.00–35.40 ^NS^	27.90–28.20	35.30–35.50	28.00–35.00
MCVr	fL	5594	92.80–112.80 **	92.50–93.30	112.50–113.10	92.00–120.00
RETH	%	5594	0.20–5.30 **	0.20–0.30	5.10–5.50	0.0–2.0
RETM	%	5594	4.00–16.90 **	3.90–4.10	16.60–17.20	2.0–11.0
RETL	%	5594	79.10–94.80 **	78.70–79.50	94.70–95.00	88.0–98.0

See ***Abbreviations and symbols*** section for the meaning of the abbreviations. Differences of means between the 2011–2020 group and general population RI (** = *p* < 0.01). ^NS^: No statistically significant difference.

**Table 5 ijerph-19-03059-t005:** Reference intervals (RIs) and basal values for total leukocytes cells and leukocytes subpopulations (females and males).

Parameter	Units	Number	2011–2020	2011–2020 90% CI		General Population RIs
				Lower Limit	Upper Limit	
WBC	10^9^/L	11,988	3.50–9.10 **	3.50–3.60	8.90–9.20	3.9–10.2
Lymphocytes	10^9^/L	11,974	1.17–3.21 **	1.16–1.19	3.18–3.24	1.1–4.5
Neutrophils	10^9^/L	11,974	1.47–5.87 **	1.44–1.49	5.73–5.99	1.5–7.7
Monocytes	10^9^/L	11,974	0.17–0.60 **	0.17–0.17	0.59–0.61	0.1–0.9
Eosinophils	10^9^/L	11,974	0.06–0.53 **	0.06–0.06	0.52–0.55	0.02–0.65
Basophils	10^9^/L	11,974	0.01–0.07 **	0.01–0.01	0.07–0.07	0.0–0.2
LUC	10^9^/L	11,974	0.06–0.25 **	0.06–0.06	0.24–0.25	0.05–1.0
Lymphocytes%	%	11,987	20.20–51.10 **	19.70–20.50	50.80–51.40	20–44
Neutrophils%	%	11,987	35.90–69.90 **	35.40–36.20	69.20–70.50	42–77
Monocytes%	%	11,987	3.50–9.00 ^NS^	3.40–3.50	8.90–9.10	2.0–9.5
Eosinophils%	%	11,986	1.00–8.80 **	1.00–1.10	8.54–9.00	0.5–5.5
Basophils%	%	11,986	0.20–1.20 **	0.20–0.20	1.10–1.20	0–1.75
LUC%	%	11,974	1.10–4.10 ^NS^	1.10–1.10	4.10–4.20	<6.5

See ***Abbreviations and symbols*** section for the meaning of the abbreviations. Differences of means between the 2011–2020 group and general population RI (** = *p* < 0.01). ^NS^: No statistically significant difference.

**Table 6 ijerph-19-03059-t006:** Reference intervals (RIs) and basal values for platelet count and associated parameters (females and males).

Parameter	Units	Number	2011–2020	2011–2020 90% CI		General Population RIs
				Lower Limit	Upper Limit	
Plt	10^9^/L	11,988	150.00–335.00 **	149.00–151.00	332.00–339.00	150–400
MPV	fL	11,988	8.30–11.60 **	8.30–8.30	11.60–11.70	5.9–9.9
Pct	%	11,975	0.15–0.31 **	0.15–0.15	0.31–0.31	0.12–0.41
PDW	%	11,975	36.50–55.10	36.40–36.60	54.70–55.40	

See ***Abbreviations and symbols*** section for the meaning of the abbreviations. Differences of means between the 2011–2020 group and general population RI (** = *p* < 0.01).

## Data Availability

The data is stored on the clinical laboratory server with restricted access. Only authorized personnel can access the clinical data.

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
