# Peer review of "Basal Values of Biochemical and Hematological Parameters in Elite Athletes"

_ijerph, 2022, doi:10.3390/ijerph19053059_

Round 1
Reviewer 1 Report
The article by Martinez et al. titled “Basal values of biochemical and hematological parameters in elite athletes” determines the reference ranges of numerous biochemical and hematological biomarkers in elite and non-elite athletes. Overall, the authors found that the basal levels of multiple biochemical and hematological biomarkers were different from reference population data, with markers of tissue damage showing the greatest difference. While some biomarkers showed increases or decreases at either the upper or lower limit of the reference range, a few biomarkers were associated with a narrow reference range compared to the reference population. This article highlights the importance of determining athlete-specific reference ranges as using normal population reference ranges can lead to problems flagging abnormal data. A major strength of this article is the large data set and analysis of all samples using the same analytical equipment and reagents. My specific comments, suggestions, and questions are listed below.
- While the article is well written, I find myself confused if the authors analyzed samples from “elite athletes” and “athletes” as indicated in lines 119-120 or if the authors analyzed 14,010 samples from 3,588 athletes. I initially understood this paragraph as the authors had data on “elite athletes” and “athletes,” however the supplemental table indicates the number of athletes = 3,588 and samples = 14,010. Could the authors provide clarification?
- If the authors collected numerous samples from a single athlete, did they control for repeat testing? If so, how?
- Lines 138-143 could be better represented as a table.
- While the authors indicate none of the female athletes were pregnant at the time of testing, did they determine if they were on contraceptives? It will also be essential to note that hormonal contraceptives can alter many biochemical and hematological biomarkers tested.
- Could the authors provide where they obtained the general population (Male and Female) RIs?
- All the conclusions for this study were based on “visual” differences between the athlete-derived RI and the general population RI. Could the authors have performed either an unpaired Student t-test was used to evaluate statistical significance for the parameters with normal distribution, or a Mann-Whitney test was used for parameters with a non-Gaussian distribution? This would strengthen the argument that athletes need a specific reference range.
- In addition to comment #6, could the same statistics be applied to the differences between male and female RIs?
- The authors provide an in-depth discussion, but the section could use more organization. For instance, could the authors provide subheadings to group specific biomarkers, such as tissue damage (CK, AST, ALT, and LDH)? This would provide needed clarity in the discussion section.
- Can the authors add the respective n(s) for all tables?
- Would you please check for grammatical errors throughout the document and the specific instances outlined below?
- Line 225-227 is a little confusing; please consider rewording.
- Line 289, There are empty parentheses.
Reviewer 2 Report
These studies provide an interesting source of knowledge about the difference between athletes and the population, but its conclusions are obvious and well documented a long time ago. As the authors themselves pointed out (line 455-461) there are factors that must be taken into account when analyzing. This, first of all, should be taken into account in the methodology of this study. This change will allow a reliable assessment of the differences. Current knowledge in the field of sports medicine and training monitoring is not based on pseudo data, but on specific analyzes. There are many articles talking about reference ranges. However, they indicate exactly the analyzed sport discipline, training or starting period. This is the direction of analyses undertaken in monitoring training. This article should therefore be more accurate in the selection of data. The general conclusions are obvious and at the same time easy to challenge due to the lack of dietary data, for example. This must be taken into account and changed in the analysis. Other comments are given below. Was the training period of athletes taken into account? Was the menstrual cycle of woman athletes taken into account? Was plasma volume evaluated? This is a key thing in the assessment. Was the dependence of the intensity and nature of the effort of these athletes evaluated? Were they different disciplines? How were athletes defined and differentiated from elite athletes? All subjects were elite athletes from different sports- Line 129. Elite or not only elite? In abstact is given differently. What kind of athletes are elite over the ages of 40 and 50? The elite concerns top athletes who compete, for example, for medals of the Olympic Games. So what kind of athletes are we talking about? Was the diet of these people analyzed? This is a significant factor affecting morphology. Were there people diagnosed with diseases (e.g. asthma, intestinal disorders) or eating disorders among the respondents? These factors significantly differentiate the results and should not be taken into account. The results should also be presented in the form of other descriptive statistics, such as deviations and medians. The average indicates only partially the result and may distort its meaning. Statistical significance should be used to assess differences in athletes' and population scores (comparisons). Current statements are not measurable. What disciplines did these people practice? The authors point to, for example, the variability of CK. Its values are higher than given in the article, e.g. for sprinters, alpine skiers. However, in endurance disciplines they are sometimes lower (up to 100 U / L after many hours of cycling) Line 240-242 – so what was the reason for this? Line 243-245 – significant changes are monitored in strength and speed disciplines, to a small extent in endurance disciplines. So how can the data from these studies be evaluated? Line 249-257- The increase in LDH is characteristic of the activation of anaerobic zones. How does this relate to the training of low- and moderate-intensity (low lactate concentration) of cyclists? As stated in the article: ,, according to their modality, the training load, aerobic/anaerobic metabolism, the lengths of their competitions, and at different stages of a competitive season in the same athletes’’ affects changes in morphology. Was this taken into account in such a large study group? Line 319-321 – what was the reason for this? Line 323-325. The article selectively refers to groups of athletes. Sometimes he talks about endurance disciplines, in other places about speed. This needs to be standardised. You cannot select part of the results according to the needs of the discussion and draw conclusions from it. Lines 341- What kind of skiers? Cross country or alpine? This is a significant difference in the aspect of research. Line 343-360. The values of these indicators are significantly influenced by diet and supplementation. Was this taken into account? 416-420- So which of these groups do the results of the article refer to? 420-428- The immune system and inflammation connect the endocrine system. The assessment of these three groups of variables gives the opportunity to correctly conclude. The most important variable in this case is cortisol. Why, then, was it omitted? 447- it is well proven that these differences exist. Kind regards.Author Response
Please see the attachment

Reviewer 3 Report
The paper presents an interesting topic regarding blood test intervals on biochemical and haematological parameters regarding the physical training, though it has serious flaws, if the authors improve their statistical analysis, the whole paper could be successfully published.
It is extremely important to identify what type of athletes- what kind of effort aerobic, anaerobic or mixed?
For example lines 47-51 -cyclists perform mostly aerobic effort- the human cardio-respiratory adaptations are well known. Values of which blood tests were assessed in that previous research?
The authors should briefly describe in the introduction what measure each biochemical and haematological parameter used in their research, and highlight the importance of these blood tests intervals in the human body adaptation in people who perform physical exercise.
The authors should introduce in limitation section that hasn't analysed the results of their research from the aerobic versus the anaerobic type of effort. Also, the authors do not offer details regarding the amount of daily training, the frequency, the years since the subjects performed regularly physical exercise.
The authors should provide abbreviations only after they have been completely spelt or defined. Please provide a description for each abbreviation used. Example- UA from Table 1
Material and methods
The authors should explain the difference between elite athletes and athletes
The authors must perform a statistical analysis test to identify the significant or non-significant differences found in their measurements compared with the general population. Therefore, their research results have no significant meaning, even are based on such an increased sample. If the authors do not have the necessary data for analysing and comparing their data with physical fitness indices, anthropometric measurements, or dietary intake, perhaps the authors should perform a comparison ( statistical analysis) based on each type of athlete included in the research, or at least between elite athletes and athletes, and why not a comparison on age groups.
Discussion- line 228- correct, but your research have included none of these factors.
Lines 233-235- the phrase should be reformulated, firstly, after statistical significance results, and if is a clear distinction regarding the types of athletes included in your research.
Many paragraphs from the discussion section should be moved to the introduction section- regarding the explanation of the importance of the biomarkers analysed. (Example lines 301-309, and others)
The author should remove the limitation section from the conclusion section, before the conclusion, as a new subchapter and highlight also the limitations regarding physical fitness indices, anthropometric measurements, or dietary intake.
The authors should rephrase lines 449-453, at least in this phase of the paper, since in the results section the authors did not provide information regarding "different stages of exercise: training, competition, to obtain the lower and the higher values".
Round 2
Reviewer 2 Report
The authors have made changes and this work is worth appreciating. However, the article still presents the obvious conclusions and the results are not properly developed. In order for it to be really useful in the current state of science, the results should be sorted by sports (endurance, speed, resistance, technical sports). This is crucial. It goes without saying that biochemical indicators change in response to exercise. The training staff of athletes judge them against discipline guidelines, not mostly reference ranges. Sports doctors are also aware of this. When presenting the results of athletes, one must not forget and not analyze the training. I appreciate the work put into the article, but it still requires some refinement. These results must be grouped and compared with each other and with the general population. All of them cannot be treated equally, as coming from athletes, because it adds nothing and says nothing. There is a large amount of error in this, and this is not what the broadening of scientific knowledge is about.
An example can be articles from e.g. 2013:
Papacosta E, Gleeson M, Nassis GP. Salivary hormones, IgA, and performance during intense training and tapering in judo athletes. J Strength Cond Res. 2013 Sep; 27 (9): 2569-80. doi: 10.1519 / JSC.0b013e31827fd85c. PMID: 23249825.
or
and
https://onlinelibrary.wiley.com/doi/abs/10.1034/j.1600-0838.2000.010002098.x
Therefore, this article should be improved by dividing the results into at least individual groups of athletes.
Reviewer 3 Report
I have no further suggestions, but please revise the references.
